# Regulation of Polyethylene Nano-Packaging on Postharvest Stipe Elongation of *Flammulina velutipes*

Yuxuan Zhao, Jianmin Yun *, Gengxin Guo, Wenhui Li, Biao Wang , Fengyun Zhao and Yang Bi

College of Food Science and Engineering, Gansu Agricultural University, Lanzhou 730070, China
* Correspondence: yunjianmin@gsau.edu.cn; Tel.: +86-138-9329-3929

**Abstract:** Stipe elongation is one of the main characteristics of the postharvest quality deterioration of *Flammulina velutipes* fruiting bodies. In order to reduce the postharvest stipe elongation of *F. velutipes* and prolong its shelf life, in this study, using polyethylene (PE) packaging and unpackaged treatments as controls, the effects of a polyethylene nano-packaging on postharvest stipe elongation of *F. velutipes* were investigated and its regulatory mechanisms were explored from the physiological and biochemical aspects. The results showed that the $CO_2$ content in polyethylene nano-packaging boxes was higher than in control boxes, and the $O_2$ content was lower than in the control group, indicating that nano-packaging could reduce *F. velutipes* respiration during low-temperature storage. The stipe elongation rate, chitinase activity, and β-1,3-glucanase activities of *F. velutipes* were lower in the nano-packaging group than in PE-packaged and unpackaged control groups, and the nano-packaging inhibited the increase in chitin and β-glucan, the main components of the cell wall. The levels of auxin (IAA), abscisic acid (ABA), gibberellin (GA), and cytokinin (CTK) were also lower in the nano-packaged group than in controls at most timepoints. After 18 d of storage, polyethylene nano-packaging inhibited the stipe elongation of fruiting bodies, and maintained mushroom quality, with stipe elongation of only 34.7% and 76.7% of PE-packaged and unpackaged control groups, respectively. The results showed that nano-packaging could effectively inhibit the harvest stipe elongation and prolong the shelf quality of *F. velutipes*.

**Keywords:** nano-packaging; *Flammulina velutipes*; stipe elongation; endogenous hormones; regulation



## 1. Introduction

*Flammulina velutipes* is a popular edible mushroom due to its good taste, high nutritional value, and various biological activities [1,2]. With the development and maturity of industrial production and cultivation technology, the yield of *F. velutipes* has increased yearly. However, due to the vigorous metabolic activity of *F. velutipes* mushrooms after harvesting, stipe elongation and fibrosis easily occur during storage, resulting in quality deterioration and great economic losses [3,4]. In order to control the postharvest deterioration and inhibit the stipe elongation of *F. velutipes* mushrooms, certain preservation methods are required. Advanced preservation technologies have been used to maintain the quality of *F. velutipes* during storage, including biological inhibitors, a modified atmosphere, and composite preservation technologies [5,6]. However, most approaches are costly, ineffective, and insecure, making them impractical for use in the food industry. Therefore, it is necessary to develop a more economical, effective, and safe preservation technology.

Nano-packaging is an emerging packaging method in the food industry. The application of nanotechnology in the field of food packaging has helped to solve the problems of food quality, safety, and stability [7]. As a new fresh-keeping method, nano-packaging has been widely used in the preservation of fruits and vegetables, such as kiwifruit, pomegranates, and strawberries [8–10]. In recent years, nano-packaging has been applied to the postharvest preservation of edible fungi. For example, Ma et al. prepared a nanocomposite film, which can better maintain the sensory quality and other characteristics

of *Agaricus bisporus* [11]. In research by Yang et al., nano-packaging was used to package *F. velutipes*, which reduced energy metabolism and prolonged the storage period [12]. Zuo et al. investigated changes in the gene expression in the *F. velutipes* transcriptome following nano-packaging, and found that nano-packaging could delay the quality deterioration of *F. velutipes* during storage [13]. However, there are few studies on the effects of nano-packaging on postharvest stipe elongation and regulation in *F. velutipes*.

For this reason, we herein explored the effects of polyethylene nano-packaging on the fruiting bodies of *F. velutipes*, including measuring changes in microenvironment gas in packaging boxes during storage, and studying the effects of nano-packaging on the stipe elongation of fruiting bodies. The effects of polyethylene nano-packaging on stipe elongation were further explored in terms of physiological and biochemical aspects, including mycelial cell wall constituents, enzymatic activities, and changes in endogenous hormone levels. The findings provided a theoretical basis and reference for maintaining the postharvest storage quality of *F. velutipes*.

## 2. Materials and Methods

### 2.1. Materials and Reagents

Fresh *F. velutipes* J5849 was harvested (white, stipe length of $15 \pm 2.0$ cm, closed veil, cap diameter of $0.8 \pm 0.2$ cm) from Shanghai Xuerong (Gansu) Biotechnology Co., Ltd. (Lintao, China), transported to the laboratory under the cold chain at 4 °C within 1 h, and with *F. velutipes* of uniform size selected.

The nano-packaging material used in this study was a polyethylene (PE) film with the addition of nano-silver particles and other chemical agents (FMSXPBC, Shanghai Fuming New Material Technology Co., Ltd., Shanghai, China). The physical properties of the nano-film were: a thickness of 35 μm; a water vapor transmission rate of $11.4$ g m$^{-2}$ 24 h$^{-1}$; carbon dioxide and oxygen transmission rates of 20,010 cm$^2$ m$^{-2}$ 24 h$^{-1}$ at 0.1 Mpa and 5659 cm$^2$ m$^{-2}$ 24 h$^{-1}$ at 0.1 Mpa; with 89% light transmittance. PE plastic wrap was produced by Tongcheng Jianxing Packaging Co., Ltd. (Tongcheng, China). Polypropylene (PP) rigid box trays (192 mm × 127 mm × 20 mm) were purchased from Beijing Linxingde Plastic Products Co., Ltd. (Beijing, China).

The main reagents, including chitin, laminarin, and β-glucan (all Sigma-Aldrich, St. Louis, MO, USA), the enzyme-linked immunosorbent assay (ELISA) kit (Shanghai Yanqi Biotechnology Ltd., Shanghai, China), and other chemical reagents, such as β-mercaptoethanol, N-acetylglucosamine, and Congo Red, were of analytical grade and produced by Tianjin Guangfu Institute of Fine Chemicals.

### 2.2. Methods

#### 2.2.1. Sample Processing

After being harvested and pre-cooled at $4 \pm 1$ °C for 24 h, fruiting bodies of *F. velutipes* were placed on 57 PP trays, with 150 g of each sample placed in each tray. All trays were randomly divided into three groups for different treatments: (I) control (CK) group, without any packaging; (II) PE packaging group, packaged with PE plastic wrap; (III) nano-packaging group, packaged with polyethylene nano-film. Each treatment included three replicates. All samples were kept in cold storage ($4 \pm 1$ °C, relative humidity 80−90%) for 18 d. Samples were taken every three days to observe and determine the relevant indicators.

#### 2.2.2. Determination of $CO_2$ Concentrations

The $CO_2$ and $O_2$ concentrations in the interior space of the package were determined by an SCY-2A $CO_2/O_2$ analyzer (Shanghai Xinrui Instrument Co., Ltd., Shanghai, China). Gas was collected by piercing the package containing the sample using a 1 mL syringe needle connected to the instrument, and the data were read immediately with the instrument. Analysis was performed in triplicate for each treatment group.

### 2.2.3. Determination of Stipe Elongation

The stipe elongation for each box of *F. velutipes* was measured at the beginning of the storage test and on each sampling day. The stipe elongation rate was calculated from the length measurement, and the value was expressed as the percentage of initial stipe elongation of fruiting bodies as follows:

$$\text{Stipe elongation rate (cm/d)} = \frac{\text{length of stipe after storage} - \text{the initial length of stipe}}{\text{storage days}}$$

### 2.2.4. Chitinase Activity Determination

According to the method of Zhang, the postharvest stipe elongation areas of *F. velutipes* (i.e., the 0−6 mm section at the top of the stipe) were selected as the test samples [14].

Chitinase was extracted according to the method of Kadoo and Badere [15]. An amount of 1 g of tissue was extracted with 2 mL of 0.1 M citrate buffer (pH 5.0), and then 300 μL of the extracted crude enzyme extract was stacked with 100 μL of sodium acetate buffer (140 mM, pH 4.5), 0.3 μmolar Sodium nitride, 1 mg of colloidal chitin, and mixed. The assay volume was adjusted to 1 mL with citrate buffer, and assays were incubated at 37 °C for 3 h. Subsequently, 0.1 mL of sodium borate buffer (0.8 M, pH 9.1) was added, and mixtures were heated in a boiling water bath for 3 min. Samples were cooled and centrifuged at $1000 \times g$ for 5 min. A 3 mL volume of dimethylamine borane (DMAB) reagent was added and incubated at 37 °C for 20 min. Subsequently, the absorbance was recorded at 585 nm, and a standard curve was plotted to quantify the amount of n-acetyl-D-glucosamine released during the incubation. The results are expressed on a fresh weight basis, and and were expressed as U kg$^{-1}$. The amount of enzyme that produces 1 μmol of N-acetylglucosamine by breaking down chitin in 1 g of tissue per hour at 37 °C is defined as one unit of enzyme activity (U).

### 2.2.5. The β-1,3-Glucanase Activity Determination

The β-1,3-glucanase activity was determined according to the method of Kadoo and Badere [15]. An amount of 1 g of tissue was extracted with 2 mL of 0.1 M citrate buffer (pH 5.0), and then 300 μL of the extracted crude enzyme extract was mixed with 480 μL of sodium acetate buffer (0.1 M, pH 5.2), and 1 mg of laminarin was mixed. The volume of the assay mixture was adjusted to 1 mL by citrate buffer, and samples were incubated at 37 °C for 3 h. Next, 500 mL of alkaline copper tartrate reagent was added and heated in a boiling water bath for 5 min. Mixtures were cooled, and 500 mL of arsenomolybdate reagent was added, followed by 3 mL of distilled water. The absorbance was recorded at 660 nm and expressed as U kg$^{-1}$; the amount of glucose released after the assay was calculated from a glucose standard curve. The amount of enzyme that produces 1 mg of reducing sugar by breaking down β-1,3-glucan in 1 g of tissue per hour at 37 °C is defined as one unit of enzyme activity (U).

### 2.2.6. Determination of Chitin Content

The method for the determination of chitin was performed as described by Kern et al., with slight modifications [16]. The deacetylation of KOH at 100 °C for 1.5 h converted chitin to chitosan, and 1.5 mL 5% (*w/v*) sodium nitrite and 1.5 mL 5% (*w/v*) potassium hydrogen sulphate were added to 1.5 mL water and mixed for 15 min. Samples were centrifuged at $1500 \times g$ for 10 min, supernatants were removed, and sulphamate salt was added and incubated for 5 min; after which, 0.5 mL of 0.5% (*w/v*) 3-methylbenzothiazol-2-one hydrazone hydrochloride was added and heated at 100 °C for 3 min. Pipes were then cooled under running cold water. Finally, 0.5 mL of 0.5% (*w/v*) ferric chloride was added, mixed, and incubated at room temperature for 30 min, and the optical density was recorded at 650 nm. Standard curves were prepared using chitin standards, and the content was expressed in mg kg$^{-1}$.

### 2.2.7. Determination of β-Glucan Content

The determination of β-glucan content was performed according to the method of Zhang et al. [17]: add 0.1 g of tissue to 1 mL of PBS buffer (pH 7.0), and centrifuge at $1000 \times g$ for 20 min to extract the supernatant. Accurately weigh 0.020 g of Congo Red and dissolve it in 0.1 mol·L$^{-1}$, pH 8.0 phosphate buffer. Dilute to 200 mL. Pipette 0.1 mL of tissue extract; add 1.9 mL of distilled water and 4.0 mL of Congo Red, in turn; and react accurately at 20 °C for 10 min. Replace the sample with 2.0 mL of distilled water as blank, and measure the absorbance at 545 nm. Standard curves were prepared using β-glucan standards, and the content was expressed in g kg$^{-1}$.

### 2.2.8. Determination of Hormones

Hormone levels were determined using ELISA kits according to the instructions. Sample pretreatment was performed as described by Meng et al. [18]. A 1 g sample of stipe tissue was mixed with 4 mL of PBS cooled to −20 °C, and the mixture was quickly ground. After centrifugation ($8000 \times g$, 4 °C) for 10 min, the supernatant was extracted. The above sample extract was purified by a C18 Solid Phase Extraction (SPE) Column. The specific steps were adding 1 mL 80% methanol to equilibrate the column, then taking 1 mL supernatant for sample extraction, and finally collecting the sample. The samples after passing through the column were blown dry with nitrogen in a water bath at 40 °C. Then, 0.1 g was taken and mixed with 1 mL PBS buffer for testing. Hormone levels were measured at 450 nm using ELISA kits according to the instructions. The results were expressed as μmol kg$^{-1}$.

### 2.3. Data Analysis

All determinations were performed at least three times, and the average value was calculated. The standard deviation of test data was calculated by Microsoft Excel 2010, analysis of variance was carried out by SPSS 18.0 ($p < 0.05$), graphs were plotted by Origin 8.0, and significance was determined by Duncan's multiple difference tests.

## 3. Results

### 3.1. Changes in Microenvironment Gas in Different Packaging Boxes during Storage

In all figures, vertical lines indicate standard error ($\pm$ SE), and different letters indicate significant differences ($p < 0.05$). All samples were kept in cold storage (4 $\pm$ 1 °C) and relative humidity (80–90%) for 18 days. PE represents those packaged with PE plastic wrap, and NA represents those packaged with PE nano-film (the same as below).

Changes in $O_2$ and $CO_2$ levels in the microenvironments of different packaging boxes during storage are shown in Figure 1. The $CO_2$ content of the two packaging groups increased rapidly during the early stages of storage, and then decreased gradually. However, the increase in $CO_2$ content in the nano-packaging group lasted longer than in the PE packaging group, and was maintained until day 9, with $CO_2$ levels as high as 11.6%, whereas an increase in $CO_2$ content in the PE packaging group could last only 3 d, with a peak $CO_2$ content of only 6.7%. Furthermore, the $CO_2$ content of the nano-packaging group was consistently higher than that of the PE packaging group in the same period. On day 9, the difference in $CO_2$ content between the two packaging groups reached a maximum, and the $CO_2$ content in the PE packaging group was only 53.0% that of the nano-packaging group. This indicated that the nano-packaging could maintain a higher $CO_2$ content in the packaging microenvironment during storage.

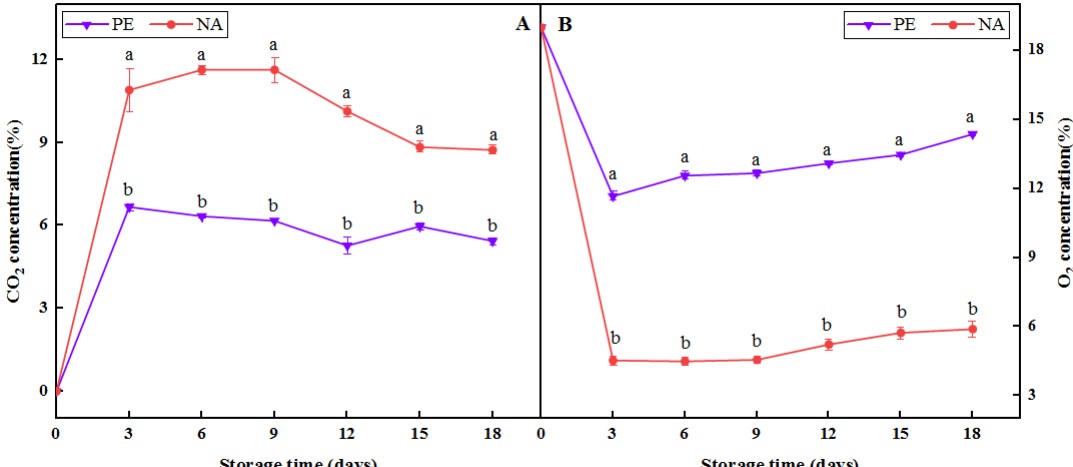

**Figure 1.** Effects of different types of packaging on $CO_2$ (**A**) and $O_2$ (**B**) levels in postharvest *F. velutipes* packaging microenvironments.

The changes in $O_2$ content in the microenvironment of different packaging groups during storage are shown in Figure 1B. The $O_2$ content of both the nano-packaging and PE packaging groups declined rapidly in the early stages, and then slowly increased and remained high during the storage period. The change lasted longer for the nano-packaging group than the PE packaging group. The changes tended to be stable; the lowest content was 4.5% at 6 d, and the content then slowly increased. In contrast, the PE group could only maintain the $O_2$ decline period for 3 d, and the lowest content of $O_2$ was 11.7%. Compared with the CK group, the $O_2$ content in the nano-packaging group was always lower than that of the PE group in the same period. On day 15 of storage, the difference between the nano-packaging group and the PE control group was the largest, and the $O_2$ content of the nano-packaging group was 42.6% that of the PE control group. This indicated that the nano-packaging could maintain a lower $O_2$ content in the packaging microenvironment during storage.

### 3.2. Effects of Nano-Packaging on the Elongation of F. velutipes Stipes

In all figures, CK represents those without any packaging (the same as below).

The elongation of stipes is one of the main manifestations of the post-ripening effect of *F. velutipes*, which makes its fruiting bodies' metabolism vigorous, and it constantly consumes its own nutrients [4]. It can be seen from Figure 2B that during the storage period, the elongation of stipes in the CK group first decreased and then increased, whereas the nano-packaging group showed a continuous upward trend. Before day 9, elongation in the CK group was lower than in the nano-packaging group, but after day 12, elongation in the CK group was higher than in the nano-packaging group. On day 6 of storage, the difference between the two groups reached a maximum, and elongation in the nano-packaging group was 1.8 times higher than in the CK group. Elongation in the PE packaging and nano-packaging groups showed a continuous upward trend during storage, and elongation in the nano-packaging group was always lower than that in the PE packaging group, and different at 3−18 d. Elongation in both groups reached a maximum on day 18 of storage, and elongation in the PE packaging group was 0.18 cm d$^{-1}$ on day 18, which was 2.9 times higher than in the nano-packaging group at this timepoint.

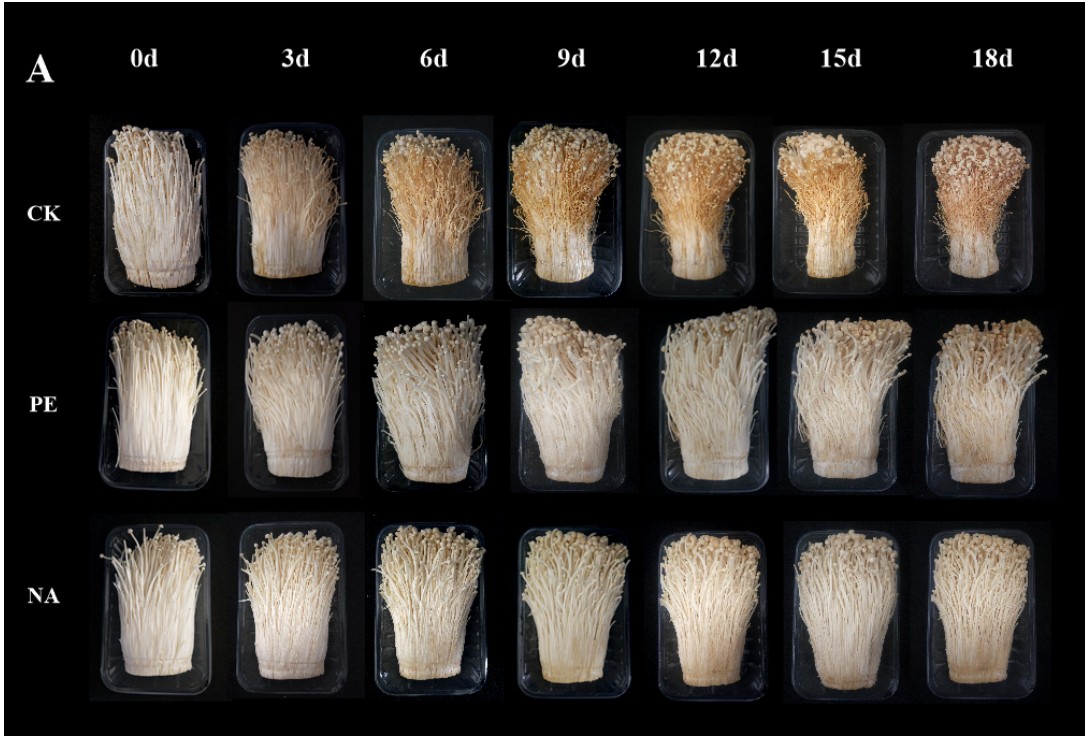

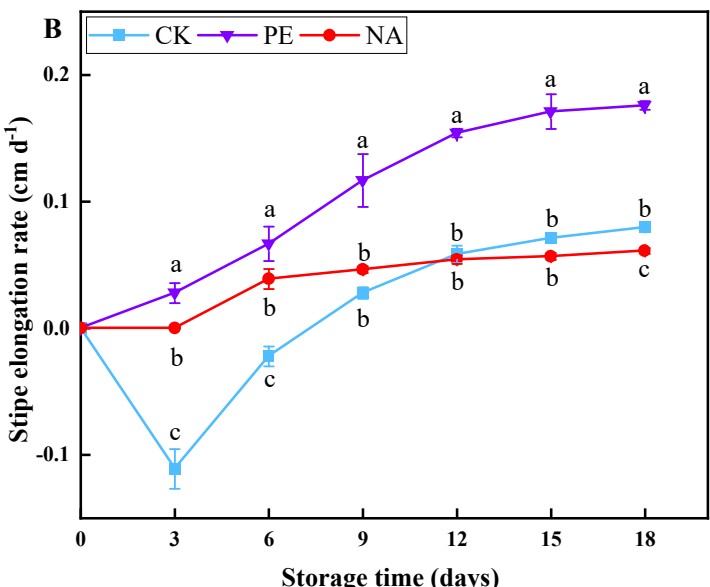

**Figure 2.** Effects of different types of packaging on sensory quality (**A**) and the postharvest elongation of *F. velutipes* stipes (**B**).

During the storage period, elongation in each packaging group showed an overall upward trend, whereas elongation in the CK group decreased in the early stage due to the shrinkage of *F. velutipes* following dehydration. During the storage period, the elongation rate of *F. velutipes* showed an upward trend, among which the nano-packaging group showed the gentlest upward trend, and the PE packaging group displayed the fastest rise, indicating that nano-packaging had an effect on inhibiting the elongation of *F. velutipes* stipe. The results of this experiment were consistent with those of Jin et al., which further confirmed that modified-atmosphere packaging can control the respiration of edible fungi

by changing the gas composition in the packaging microenvironment, and thereby reduce the consumption of nutrients and maintain storage quality [19].

### 3.3. Effects of PE Nano-Packaging on the Cell Wall of F. velutipes Stipes

#### 3.3.1. Chitinase Activity

Chitinase is present in the stipe cells of macrofungi, where it plays a key role in the elongation of the stipe cell wall [20]. During storage, chitinase activity in the stipes of *F. velutipes* displayed an upward trend in all three groups, and reached a maximum value on day 18 (Figure 3). The chitinase activity of the nano-packaging group was lower than that of the CK group and the PE packaging group from day 6 to day 18. On day 18, the gap between the nano-packaging group and the other two groups was the largest (66.2% and 58.5% of CK and PE packaging groups, respectively). These results indicated that nano-packaging inhibited the chitinase activity in *F. velutipes* stipes.

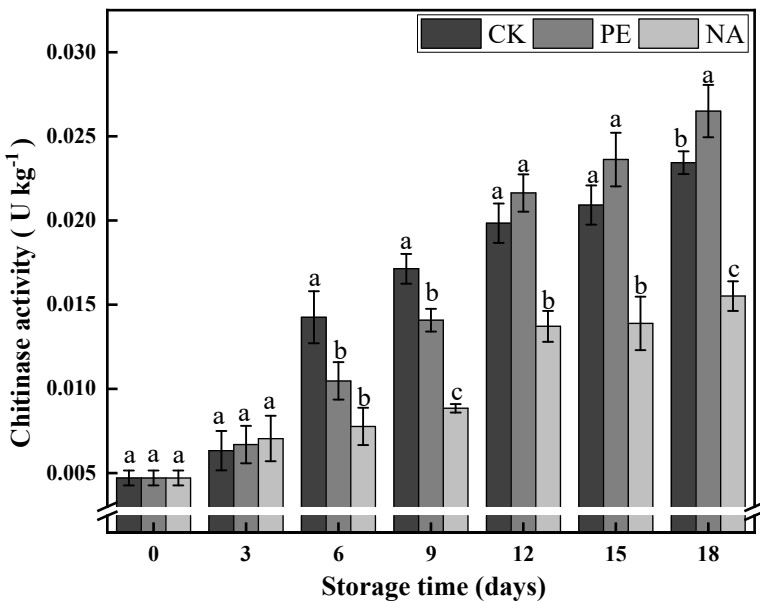

**Figure 3.** Effects of different types of packaging on chitinase activity in the elongation zone of postharvest *F. velutipes* stipes. Vertical lines indicate standard error ($\pm$ SE), and different lower-case letters a, b and c indicate significant differences ($p < 0.05$). All samples were kept in cold storage ($4 \pm 1\ ^{\circ}$C) and relative humidity (80–90%) for 18 days. CK represents without any packaging, PE represents those packaged with PE plastic wrap, and NA represents those packaged with PE nano-film (the same as below).

#### 3.3.2. β-1,3-Glucanase Activity

β-glucanase plays a role in regulating the stipe wall, including wall remodelling, and cooperates with chitinase to induce stipe wall extension [20]. As shown in Figure 4, the β-1,3-glucanase enzyme activity in the stipe cell wall initially increased and then leveled-off during storage in all three groups. However, β-1,3-glucanase activity in the nano-packaging group was lower than in the CK and PE packaging groups from day 9 to day 18. On day 15, the gap between nano-packaging and CK groups was the largest (65.8% of the CK group). On day 18, the gap between nano-packaging and PE packaging groups was the largest (73.5% of PE packaging). These results showed that nano-packaging inhibited β-1,3-glucanase enzyme activity in the cell wall of *F. velutipes* stipes.

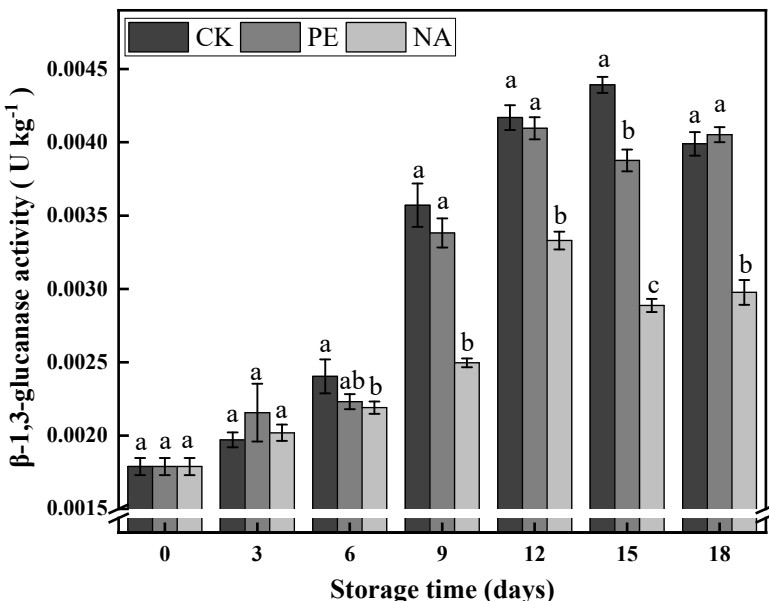

**Figure 4.** Effects of different types of packaging on β-1,3-glucanase activity in the elongation zone of postharvest *F. velutipes* stipes.

### 3.3.3. Chitin Content

Chitin is an important component of the cell wall of *F. velutipes* stipes, and an increase in chitin content indicates stipe elongation [20]. It can be seen from Figure 5 that the chitin content of *F. velutipes* stipes displayed a rising trend from day 0 to day 18 for all three groups. However, compared with the other two groups, the rise in the nano-packaging group was gentler, and the chitin content of the nano-packaging was also lower than that of the CK and PE packaging groups. On day 15 of storage, the chitin content of the CK group was 1.76 times that of the nano-packaging group; on day 18, the chitin content of the PE packaging group was 1.83 times higher than that of the nano-packaging group. Thus, nano-packaging inhibited the increase in chitin content during 0−18 d of storage.

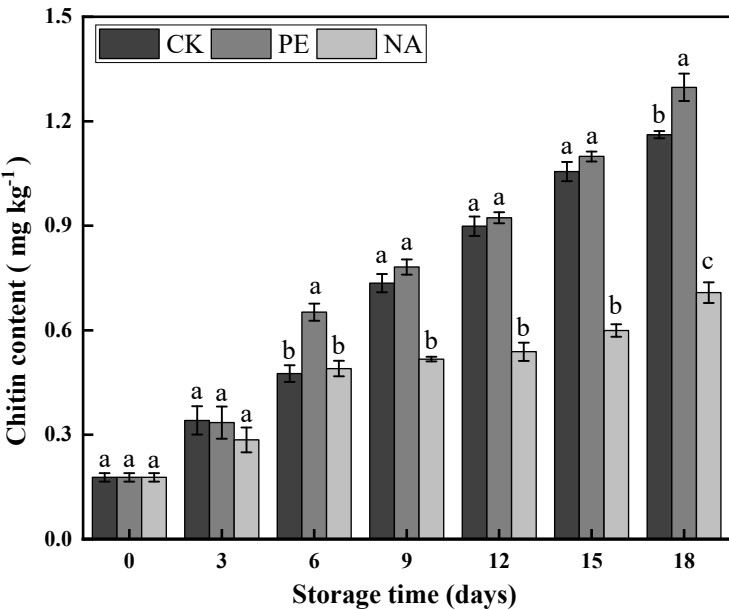

**Figure 5.** Effects of different types of packaging on chitin content in stipe elongation area of *F. velutipes*.

### 3.3.4. β-Glucan Content

β-glucan is also a major component of the fungal cell wall, which, together with chitin, constitutes the structural scaffold of the cell wall [20]. It can be seen from Figure 6 that the β-glucan content of the three groups displayed a continuously increasing trend during the storage period, but the increase in the nano-packaging group was the slowest. During 6−18 d, the β-glucan content was always lower in the nano-packaging group than the other two groups; on day 18 of storage, the β-glucan content of the nano-packaging group was 80.1% and 64.3% that of the CK and PE packaging groups, respectively. This suggests that nano-packaging had an inhibitory effect on the increase in β-glucan content during storage.

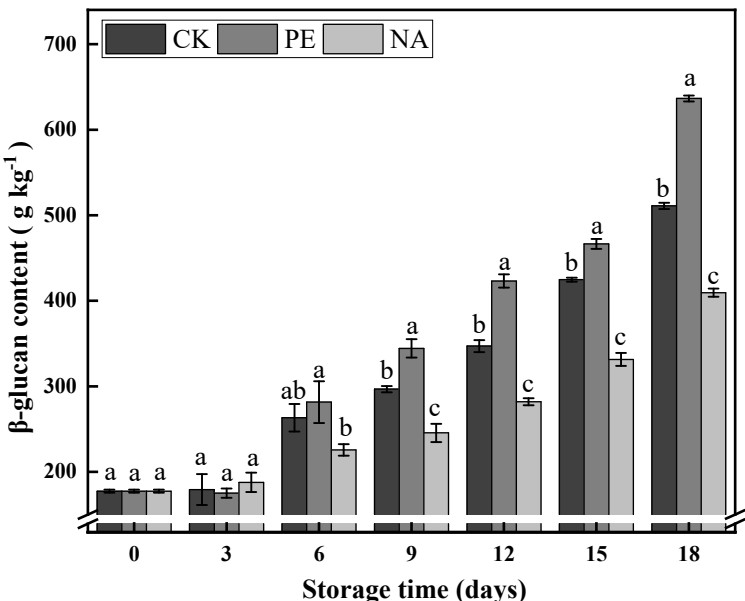

**Figure 6.** Effects of different types of packaging on the content of β-glucan in the elongation area of *F. velutipes* stipes.

### 3.4. Effects of Polyethylene Nano-Packaging on Endogenous Hormones in F. velutipes

Indole-3-acetic acid (IAA) is an endogenous growth hormone that plays an important role in cell elongation in plant hypocotyls, epicotyls, and other organs [21,22]. According to Meng's research, IAA also appeared in *Agaricus bisporus* [18]. According to research reports, the exogenous addition of IAA can increase the biomass production of the edible mushroom, *Pleurotus sajor-caju* [23]. IAA affects cell elongation by regulating elongation of the cell wall [24]. It can be seen from Figure 7A that the IAA content of *F. velutipes* first decreased and then increased in all three groups during the storage period, and it reached the lowest value at 9−12 d. Compared with CK, the IAA content of PE and nano-packaging groups decreased more significantly during 3−6 d, but the increase in IAA content in the nano-packaging group was slower during the rising period (12−18 d). When stored for 18 d, the IAA content of the nano-packaging group was lower than the other two groups. The gap between the nano-packaging and CK groups reached the maximum on day 6 (88.9% that of the CK group). The gap with the PE packaging group reached a maximum on day 18 (90.8% that of the PE packaging group). Thus, nano-packaging had a clear inhibitory effect on the IAA content of the stipes of *F. velutipes* after harvest.

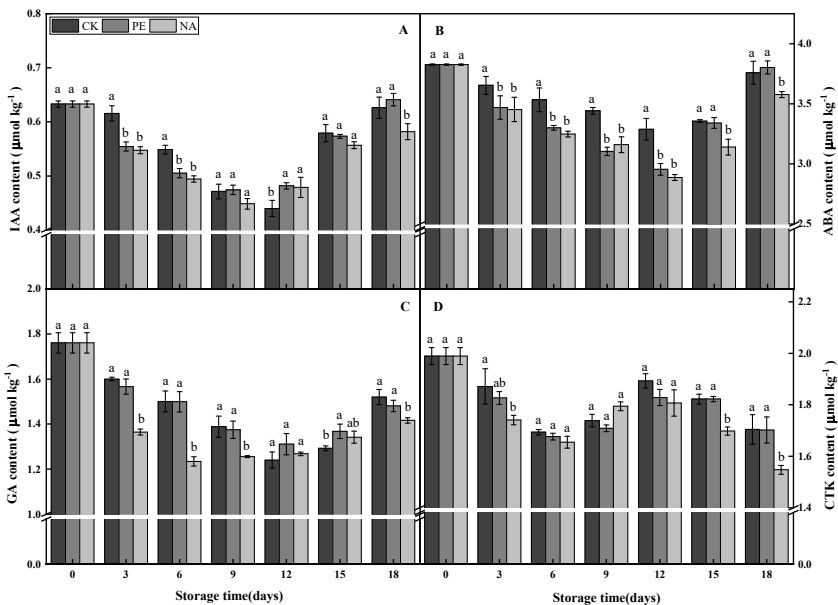

**Figure 7.** Effects of different types of packaging on IAA (**A**), ABA (**B**), GA (**C**), and CTK (**D**) levels.

Abscisic acid (ABA) is an endogenous plant hormone that also affects cell elongation, mainly by inducing lateral elongation [24]. In Serpil et al., ABA was present in *Pleurotus florida* (Basidiomycetes), and the growth rate of this fungus was negatively correlated with ABA synthesis [25]. In recent years, studies have shown that ABA also exists in the fungi of basidiomycetes, such as *Volvariella volvacea* and *Agaricus bisporus* [18,26]. It can be seen from Figure 7B that the ABA content of *F. velutipes* first decreased and then increased during 18 d of storage in all three groups. However, the ABA content of the nano-packaging group was always lower than that of the CK group, and on day 12 of storage, the gap between the nano-packaging and CK groups reached a maximum (87.8% that of the CK group). Moreover, although there was no difference in ABA content between nano-packaging and PE packaging groups in the early storage period (0−15 d), the ABA content of the nano-packaging group was also lower than the PE packaging group in the later storage period (15−18 d), and on day 18, the gap between the nano-packaging and PE packaging groups was the largest (94.1% that of the PE packaging group). This indicated that nano-packaging inhibited ABA production in the stipes of *F. velutipes* during storage.

Previous studies have shown that gibberellin (GA) plays a key role in stem elongation [27–29]. GA is also present in basidiomycetes [18,26]. According to previous studies, exogenous GA3 treatment can maintain the postharvest quality of *Agaricus bisporus* [30]. GA can act on the elongation of cells, and, like IAA, especially the longitudinal elongation of cells [24]. The GA content of *F. velutipes* after harvest first decreased and then increased during storage (Figure 7C). Compared with the other two groups, the GA content of the nano-packaging group was lower after 18 d of storage. On day 6, the gap between the nano-packaging group and the other two groups reached a maximum (82.3% and 82.4% that of CK and PE packaging groups, respectively). These results showed that nano-packaging inhibited the GA content of *F. velutipes* stipes after harvest.

Plants, fungi, and bacteria can produce cytokinin (CTK), and its functional roles in fungi and prokaryotes are unclear, but CTK can induce plant cells to divide and promote cell growth [31]. The CTK content of *F. velutipes* after harvest during 18 d of storage first decreased, then increased, and then decreased again (Figure 7D). All three groups displayed a downward trend at 0−6 d and reached a maximum at 12 d, which then showed a downward trend. The content of CTK in the nano-packaging group was not different from the other two groups in the early stages of storage (0−9 d), but it was lower than the other two groups in the later stages of storage (15−18 d). On day 18, the CTK content of the nano-packaging group reached a maximum difference from CK and PE packaging groups

of 90.8% and 90.9%, respectively. This indicated that nano-packaging inhibited the growth of stipes by regulating the synthesis of CTK in stipe cells in the later stages of storage.

## 4. Discussion

Stipe elongation is one of the most significant characteristics of fruiting body growth and development in basidiomycetes mushrooms. However, the phenomenon of stipe elongation occurring in postharvest *F. velutipes* mushrooms is considered to be the beginning of aging and the gradual loss of commercial value [20,32]. Studies have shown that stipe cells are enclosed in the cell wall and provide sufficient strength for the cell wall to withstand turgor during the elongation and growth of the stipe, while maintaining sufficient plasticity to extend the cell wall under turgor [20,33]. In *F. velutipes*, stipe elongation is limited to the first few millimetres of the apical region of the stipe [20,34]. Research showed that postharvest stipe elongation in *F. velutipes* occurred at $0-6$ mm from the top, and elongation was more obvious at $0-2$ mm [14]. Therefore, in this experiment, the elongation area at the top of the fruiting body stipe of *F. velutipes* was collected as sample material to investigate the effects of different types of packaging on stipe elongation, and to explore its regulation mechanism during storage. The results of this study showed that compared with the control group, the stipe elongation of *F. velutipes* treated with nano-packaging was effectively inhibited during the whole storage period. This indicated that nano-packaging played an effective role in inhibiting the stipe elongation of *F. velutipes*.

Almost all fungi use chitin, β-1,3-glucan, and various glycoproteins as major components of their cell walls [20,35,36]. Previous studies have reported that the activities of chitinase and β-1,3-glucanase are important factors affecting fungal stipe elongation. For example, Mol and Wessels observed that the elongated apical stipe cell wall of *Agaricus bisporus* was more susceptible to chitinase and extracellular β-1,3-glucanases than the non-elongated basal stipe cell wall [20]. Other research suggested that chitinase played a key role in the elongation and growth of stipes [35]. Although β-1,3-glucanase alone cannot completely loosen the cell wall to maintain continuous stipe wall extension, it can synergize with low concentrations of chitinase to induce stipe cell wall extension [37]. The degradation of chitin by chitinase results in the separation of chitin microfibrils from each other to increase the insertion of newly synthesised chitin and matrix polysaccharides into the wall space under turgor to facilitate stipe elongation growth [38]. The results of the present work showed that nano-packaging could inhibit the activities of chitinase and β-1,3-glucanase in *F. velutipes*, and slow the increase in chitin and β-glucan content. According to previous studies, nano-packaging can maintain lower membrane permeability and oxidation degree, thereby delaying the deterioration of *F. velutipes* [39]. Some previous studies showed that higher $CO_2$ and lower $O_2$ in packaging can inhibit the respiration of *F. velutipes* and reduce the anabolism of stipe elongation, thereby inhibiting the elongation of stipes [14]. The results of this study are similar to the above conclusion. Since a high concentration of $CO_2$ inhibits the respiration of *F. velutipes*, affecting energy metabolism and ATP supply, it also inhibits the activity of various enzymes related to energy metabolism, such as tricarboxylic acid cycle (TCA) and its corresponding enzymes [40,41]. Chitin and β-1,3-glucan in the cell wall are carbohydrates, and their synthesis requires the support of TCA and its related enzymes. As such, the high-$CO_2$ and low-$O_2$ environment under the nano-packaging can inhibit the content of chitin and β-1,3-glucan and the corresponding enzyme activities in the stipe cell wall of *F. velutipes*.

Studies have shown that the main factors causing stipe elongation in *F. velutipes* were cell wall synthesis and cell elongation. In order to expand, fungal cells must break free from the restrictive cell wall, so that cell elongation is accompanied by the expansion of the cell wall [42]. Stipe elongation growth is primarily attributable to multiple cell elongations rather than an increase in the number of cells [20,43]. According to Kende and Zeevaart, hormones can affect cell elongation [24]. For example, the effects of growth regulators, such as IAA, ABA, GA, and CTK, on macrofungi may be due to their induction of cell division and elongation. GA stimulates the elongation of stem tissue, and ABA is an aging hormone

that stimulates the maturation or aging of the body. Our research showed that the levels of ABA in the elongation area of *F. velutipes* stipes were higher than others, reaching 3.83 μmol $kg^{-1}$. This suggested that ABA regulated cell growth and aging in the elongation area of *F. velutipes* stipes. At the same time, the ABA level of *F. velutipes* under the nano-packaging was lower than that of the control group, and it was more obvious at 15–18 d. This indicated that the nano-packaging inhibited the regulatory effect of ABA on *F. velutipes*. In previous studies, IAA and GA were shown to have synergistic effects in tobacco, barley, and *Agaricus bisporus* [18]. In the present study, the content of GA under nano-packaging was lower than that of the control group during storage, whereas the content of IAA did not change significantly, and was only inhibited at the end of storage at 18 days compared with the control. However, the IAA and GA levels were similar; hence, we speculated that they play a coordinated role in the elongation of stipe cells in *F. velutipes* after harvest. The CTK content of the nano-packaging group was lower than that of the other two groups at 15–18 d in the late storage period, which indicated that it regulated the synthesis of CTK in the stipe cells in the late storage period. Some previous studies also showed that the rapid elongation of stipes is due to the elongation of stipe cells rather than cell division [26], which has little to do with CTK, because CTK mainly promotes cell division [31]. There was little change in CTK content in the stipe throughout the storage period, consistent with its insignificant regulatory influence on *F. velutipes* stipes. Therefore, we concluded that nano-packaging could regulate the level of endogenous hormones in postharvest *F. velutipes*, thereby inhibiting the elongation and division of cells in the elongation zone of the stipe. As mentioned above, under the environment of high $CO_2$ and low $O_2$, the synthesis of various enzymes is affected by inhibiting energy metabolism [41]. As such, we speculate that the high-$CO_2$ and low-$O_2$ environment under nano-packaging also affects the related enzyme activities and metabolic pathways of endogenous hormone synthesis by inhibiting respiration to regulate ATP energy charge levels, thereby regulating hormone levels.

## 5. Conclusions

In this study, the content of components related to cell division and growth, the activities of related enzymes, and the levels of endogenous hormones in the stipe elongation area of *F. velutipes* stored at a low temperature ($4 \pm 1$ °C) for 18 d were measured, and the effects of nano-packaging on stipe elongation were explored. The results showed that nano-packaging maintained high $CO_2$ and low $O_2$ levels in the packaging bag, and reduced the activities of chitinase and β-1,3-glucanase, thereby decreasing the increase of chitin and β-glucan content. It also lowered the abundance of endogenous hormones in the stipe elongation area, and inhibited the elongation and division of stipe cells, and thereby inhibited stipe elongation of the fruiting bodies of *F. velutipes*.

**Author Contributions:** Conceptualization, Y.Z. and J.Y.; methodology, Y.Z. and G.G.; software, W.L. and Y.Z.; validation, Y.B. and J.Y.; formal analysis, Y.Z.; investigation, Y.Z. and B.W.; resources, J.Y.; datacuration, Y.Z.; writing—original draft preparation, Y.Z.; writing—review and editing, Y.Z. and J.Y.; visualization, Y.Z., Y.B., and F.Z.; supervision, J.Y.; project administration, Y.B.; funding acquisition, J.Y. All authors have read and agreed to the published version of the manuscript.

**Funding:** This research was funded by the National Key R&D Program of China, grant number 2018YFD0400205.

**Data Availability Statement:** The authors confirm that the data supporting the findings of this study are available within the article.

**Acknowledgments:** We thank International Science Editing for editing this manuscript (http://www.internationalscienceediting.com) (accessed on 1 June 2022).

**Conflicts of Interest:** The authors declare no conflict of interest.

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
