# Peer review of "Regulation of Polyethylene Nano-Packaging on Postharvest Stipe Elongation of Flammulina velutipes"

_agronomy, doi:10.3390/agronomy12102362_

Round 1
Reviewer 1 Report
The manuscript describes the results and does not answer the reason. The discussion is not based on the treatments studied (type of packaging). The manuscript does not characterize and define in detail what polyethylene nano-packaging has to present superior results to PE-packaged and unpackaged control groups.
Author Response
1.Moderate English changes required.
Response to comment: First of all, we thank the reviewer for his rigor and meticulousness. We agree with his comment and have made corresponding correction in the manuscript.
2.The introduction should provide sufficient background and include all relevant references.
Response to comment: As the reviewer’s suggestion, we have improved the research background in the introduction, and added some relevant references. The revisions have been marked in blue in the paper.
3.The methods should be described adequately.
Response to comment: We greatly appreciate the reviewer's suggestion. we have made further additions in the Inadequately Described Methods section, which has been marked in blue.
4.The results should be presented clearly.
Response to comment: According to the reviewer's suggestion, we have supplemented the Results section, which has been marked in blue.
5.All the cited references relevant to the research, research design, and conclusions supported by the results should be improved.
Response to comment: We thank for the reviewer's suggestion. We have improved some of the cited references related to the study, study design, and conclusions supported by the results,which has been marked in blue.
Reviewer 2 Report
In this paper, regulation of polyethylene nano-packaging on postharvest stipe elongation of Flammulina velutipes were studied. The findings provided a theoretical basis and reference for maintaining the post-harvest storage quality of F. velutipes. This work is novel and the story is clear. However, before the manuscript can be accepted for publication, some minor revisions are required.
1. The authors should introduce the research background at the beginning of the abstract and summarize the conclusion or significance of the research at the end of the abstract.
2. The keywords section should be added " regulatory mechanism"..
3. In line 76, the Latin " F. velutipes " has a writing format problem, which should be changed with italic.
4. In line 89, the word "Congo rRed" is a writing error. Please check and modify it carefully.
5. Line 82, the abbreviation "PE" that appears for the first time in article should be written in full
6. “3. Results and Discussion” should be changed to “3. Results”.
7. In lines 221-224, the relationship between nutrient content and stalk elongation needs to be described in detail.
8. In lines 326-327, it would be better to introduce the importance of studying stalk elongation, such as its effect on the quality of Flammulina velutipes.
9. The Figure legends should be self-explanatory. Please include the description of the CK, PE, and NA at least in one figure legend and refer to this description in the other figures as "See Figure 1 or Figure 2 for treatment description". In Figure 1, CK treatment does not appear, please check this description to the legend of Figure 1
Author Response
1.The authors should introduce the research background at the beginning of the abstract and summarize the conclusion or significance of the research at the end of the abstract.
Response to comment: We thank the reviewer for his rigor and meticulousness, we have added content according to the reviewer’s suggestion. For details, please refer to the revised text, which has been marked in red.
2.The keywords section should be added " regulatory mechanism".
Response to comment: As the reviewer’s suggestion, we have added " regulatory mechanism" in the keywords section, which has been marked in red.
3.In line 76, the Latin " F. velutipes " has a writing format problem, which should be changed with italic.
Response to comment: We would like to thank the reviewer for his rigorous attitude, we have corrected it. For details, please refer to the revised text, which has been marked in red.
4.In line 89, the word "Congo rRed" is a writing error. Please check and modify it carefully.
Response to comment: We would like to thank the reviewer for his rigorous attitude. we have corrected it according to the reviewer’s suggestion. For details, please refer to the revised text, which has been marked in red.
5.Line 82, the abbreviation "PE" that appears for the first time in article should be written in full
Response to comment: We would like to thank the reviewer for his rigorous attitude. We are very sorry that due to our negligence. We found that the abbreviation “PE” that appears for the first time in line 77 and have written it in full. For details, please refer to the revised text, which has been marked in red.
6.“3. Results and Discussion” should be changed to “3. Results”.
Response to comment: We thank for the reviewer's suggestion. We have changed “3. Results and Discussion” to “3. Results” of the manuscript. For details, please refer to the revised text, which has been marked in red.
7.In lines 221-224, the relationship between nutrient content and stalk elongation needs to be described in detail.
Response to comment: We thank for the reviewer's suggestion. We have refined the description on the relationship between nutrient content and stipe elongation in lines 230-232. For details, please refer to the revised text, which has been marked in red.
8.In lines 326-327, it would be better to introduce the importance of studying stalk elongation, such as its effect on the quality of Flammulina velutipes.
Response to comment: We thank for the reviewer's suggestion. We have added “the importance of studying stipe elongation” of the manuscript. For details, please refer to the revised text, which has been marked in red.
9.The Figure legends should be self-explanatory. Please include the description of the CK, PE, and NA at least in one figure legend and refer to this description in the other figures as "See Figure 1 or Figure 2 for treatment description". In Figure 1, CK treatment does not appear, please check this description to the legend of Figure 1
Response to comment: We have corrected it according to the reviewer’s suggestion. For details, please refer to the revised manuscript and figure 2, which has been marked in red.
Reviewer 3 Report
Dear colleagues,
The manuscript seems me interesting and new.
However, I have a number of questions and comments.
1. It is not clear why this nano-packaing material was chosen?
2. What was taken as a unit of enzyme activity (U)?
3. How hormones were identified? Part 2.2.8 is not clear. What is column C18? Is this an HPLC column?
4. What do the small letters a and b in figure 1 mean?
5. Explain lines 185-186. Figure 1B shows the dynamics of O2?
6. In my opinion, figures 2A and 2B should be divided into two or 2A should be moved to Supplemented Materials.
7. Parts 3.2 and 3.3 have the same title, although 3.3 describes enzymatic activity.
8. What can cause inhibition of enzyme activity? Is it a change in the CO2/O2 ratio or an effect of the nano-material itself? Perhaps this is the toxicity of the nanomaterial?
9. The same applies to the inhibition of chitin and glucan.
10. Part 3 titled "Results and discussion". Then separately part 4 "Discussion". 11. The authors write that IAA, ABA, GA are plant hormones and determine them in the fungus. What is known about the production of these hormones by basidiomycetes? According to CTK that it is produced by fungi, plants and bacteria, so it is understandable.
12. Why is there regulation? Is it the effect of the material itself or dense packaging in a film? What happens if you use a different film? Some kind of hypothesis is needed here.
Author Response
1.It is not clear why this nano-packaing material was chosen?
Response to comment: We greatly appreciate the reviewer's comment. As we mentioned in the introduction of this paper, nano-packaging is an emerging packaging method in the food industry. The reason why we choose this kind of nanofilm to carry out our research is according to the fresh-keeping needs of mushrooms, based on the composition and series of physical properties of this kind of nanofilm (including thickness, water vapour transmission rate, carbon dioxide and oxygen transmission rates and light transmittance) , and selected it on the basis of the comparison of several cling film experiments . Meanwhile, due to the length of the article, we did not present the content of the previous comparative experiments in this article.
2.What was taken as a unit of enzyme activity (U)?
Response to comment: According to the reviewer's comments, we have made additional clarifications in the methods section of the enzymatic activity assay.
3.How hormones were identified? Part 2.2.8 is not clear. What is column C18? Is this an HPLC column?
Response to comment: Based on the reviewer's comments, we have supplemented the detailed method in the hormone assay section (Part 2.2.8). which has been marked in purple. “C18 column” refers to C18 Solid Phase Extraction Column, which is a sample pretreatment device developed from chromatography column for extraction, separation and concentration, which is different from HPLC column. The specific name of the C18 column has been modified in the manuscripts, which has been marked in purple.
4.What do the small letters a and b in figure 1 mean?
Response to comment: Small letters a and b in Figure 1 represent the significance of differences among different treatments on the day, and different letters represent significant differences.
5.Explain lines 185-186. Figure 1B shows the dynamics of O2?
Response to comment: We would like to thank the reviewer for his rigorous attitude. In line186, we added an extra B after Figure 1 due to our carelessness. The error has now been corrected and marked in purple. Yes, Figure 1 B shows the dynamics of O2.
7.In my opinion, figures 2A and 2B should be divided into two or 2A should be moved to Supplemented Materials.
Response to comment: We thank for the reviewer's suggestion. Yes,You are right. In fact, Figures 2A and 2B illustrate the changes in the stipe elongation of F. velutipes from two different aspects of sensory and data. So, we think that putting the figures 2A and 2B together can make the description more sufficient.
7.Parts 3.2 and 3.3 have the same title, although 3.3 describes enzymatic activity.
Response to comment: We agree with the reviewer's comment. We have revised the title of 3.3 to “Effects of PE nano-packaging on the cell wall of F. velutipes stipes” , which has been marked in purple.
8.What can cause inhibition of enzyme activity? Is it a change in the CO2/O2 ratio or an effect of the nano-material itself? Perhaps this is the toxicity of the nanomaterial?
Response to comment: We would like to thank the reviewer for his rigorous attitude. Regarding the reasons for the inhibition of enzymatic activity, we have supplemented the analysis in the Discussion section of the text. See the revised manuscript for details, and the revisions are marked in purple. In addition, the nanomaterials packaging product selected in this study is a food-grade packaging material that has been tested for safety before leaving the factory. Therefore, it is clear that it is not the toxicity of the nanomaterial itself.
9.The same applies to the inhibition of chitin and glucan.
Response to comment: The answer to this question is the same as the previous question 8. We have supplemented the analysis in Discussion section of the text. See the revised manuscript for details, and the revisions are marked in purple.
10.Part 3 titled "Results and discussion". Then separately part 4 "Discussion".
Response to comment: We thank for the reviewer's suggestion. We have changed “3. Results and Discussion” to “3. Results” of the manuscript, which has been marked in red.
11.The authors write that IAA, ABA, GA are plant hormones and determine them in the fungus. What is known about the production of these hormones by basidiomycetes? According to CTK that it is produced by fungi, plants and bacteria, so it is understandable.
Response to comment: We greatly appreciate the reviewer's comment.Up to now, the researches on the endogenous hormones IAA, ABA and GA are mainly concentrated in the field of plant, and there are few reports on the production of these endogenous hormones by basidiomycetes. However, there are also reports in the literature that basidiomycetes also produce these hormones. For example, according to reference 18, Meng's research showed that endogenous hormones IAA, ABA and GA existed in Agaricus bisporus, and changes in their content also have regulatory effects on the stipe and gills. According to reference 24, Xie's research showed that there were endogenous hormones ABA and GA in Volvariella volvacea, and GA had a regulatory effect on the elongation of its stipe. It may be due to the inaccuracy and rigor of our statements in the manuscript that the reviewers have such doubts. It has been revised and improved, please refer to the manuscript for the revised content, which has been marked in purple.
12.Why is there regulation? Is it the effect of the material itself or dense packaging in a film? What happens if you use a different film? Some kind of hypothesis is needed here.
Response to comment: We would like to thank the reviewer for his questions. The regulation here refers to the action of aging regulating of F. velutipes mushrooms by different CO2/O2 ratios in different packages. The reason for the effect is the difference in gas content in the packaging environment caused by the different oxygen permeability of different materials. For example, Zhang used several different packagings in the previous experiments in the laboratory, and found that the CO2/O2 ratios in different packaging environments were different. At the same time, under the conditions of high CO2 and low O2 in the two packaging microenvironments, the stipe elongation of F. velutipes was better restrained. In this present work, PE packaging are also compared with nanomembranes, and the results are similar to those of the above studies.
Round 2
Reviewer 3 Report
Dear colleagues! After the correction, the manuscript has been improved significantly. I have two small remarks. (1) In my opinion, the "regulatory mechanism" should be removed from the Keywords, because the article does not study regulatory mechanisms. The term "regulation" is enough. References 18 and 24 are not available to the reader because they are in Chinese. Can you replace them? This is also my personal interest, because I ask the information on the production of hormones by basidiomycetes for my research.
Author Response
1.In my opinion, the "regulatory mechanism" should be removed from the Keywords, because the article does not study regulatory mechanisms. The term "regulation" is enough.
Response to comment: As the reviewer’s suggestion, we have removed " regulatory mechanism" in the keywords section.
2.References 18 and 24 are not available to the reader because they are in Chinese. Can you replace them? This is also my personal interest, because I ask the information on the production of hormones by basidiomycetes for my research.
Response to comment: According to the reviewer's comments, we have added the foreign literature on the relationship between endogenous hormones and mushrooms. At the same time, we have added new explanations in the results section, which has been marked in green.